# Place cells are more strongly tied to landmarks in deep than in superficial CA1

Tristan Geiller[1,2], Mohammad Fattahi[1,3], June-Seek Choi[2] & Sébastien Royer[1,3]

Environmental cues affect place cells responses, but whether this information is integrated versus segregated in distinct hippocampal cell populations is unclear. Here, we show that, in mice running on a treadmill enriched with visual-tactile landmarks, place cells are more strongly controlled by landmark-associated sensory inputs in deeper regions of CA1 pyramidal layer (CA1d). Many cells in CA1d display several firing fields correlated with landmarks, mapping positions slightly before or within the landmarks. Supporting direct involvement of sensory inputs, their firing fields show instantaneous responses to landmark manipulations, persist through change of context, and encode landmark identity and saliency. In contrast, cells located superficially in the pyramidal layer have single firing fields, are context specific and respond with slow dynamics to landmark manipulations. These findings suggest parallel and anatomically segregated circuits within CA1 pyramidal layer, with variable ties to landmarks, allowing flexible representation of spatial and non-spatial information.

[1] Center for Functional Connectomics, Korea Institute of Science and Technology, Seoul 136-791, Republic of Korea. [2] Department of Psychology, Korea University, Seoul 136-701, Republic of Korea. [3] Department of Neuroscience, Korea University of Science and Technology, Daejeon 305-350, Republic of Korea. Correspondence and requests for materials should be addressed to S.R. (email: royers@kist.re.kr).

Environmental cues play a prominent role in the implementation of hippocampal place cells, with the manipulation of maze walls and objects inducing the reconfiguration or remapping of place fields[1–5]. Yet, place cells are not tied only to environmental cues, but are also controlled by factors such as travel distance, speed, goal, time and memory[6–10]. To what extent this diverse information is integrated versus segregated in distinct hippocampal cells populations is unclear. To date, place cells have been generally investigated as a single mechanism within a given CA region. However, in the CA1 region particularly, the anatomical data suggest that several mechanisms might be present and segregated.

First, different information reaches CA1 through segregated pathways and target specific CA1 sub-regions. Non-spatial information from the lateral entorhinal cortex (LEC)[11–16] and spatial information from the medial entorhinal cortex (MEC)[17,18] target the proximal and distal regions of CA1, respectively[19,20], underlying differences in place field tuning along the proximodistal axis[11,21]. And along the radial axis of CA1 pyramidal layer, the deep layer (CA1d, bordering oriens) receives about 2.5 times more CA2 inputs than the superficial layer (CA1s, bordering radiatum)[22]. This comes in addition to differences in local circuits, molecular expression[23] and physiological properties, with notably CA1d and CA1s pyramidal cells showing differences in number of place fields, bursting activity, spike phase relationship with theta/gamma oscillations[24], reward influence[25] and firing activity during ripples oscillations[26,27]. Second, CA1 intrinsic connectivity is well suited for functional division, compared with CA3 for instance. The CA3 network is highly recurrent, with CA3-to-CA3 inputs largely outnumbering inputs from the entorhinal cortex and dentate gyrus[20]. In contrast, the CA1 network is mainly a feedforward network with almost no inter-connections between pyramidal cells, allowing cell groups to behave independently and even to compete via feed-forward inhibition[28]. Accordingly, when a subset of environmental cues is moved, cells in CA1 split in two groups, in line with the altered and the stationary cues[5], while CA3 cells respond in a coherent manner.

Place cells are typically studied in open arena and maze environments rich with visual cues (maze/room cues, walls, corners), which can pose a problem for discerning place field mechanisms. For example, cells called landmark-vector cells (LV cells) display several place fields correlated with the position of objects in maze, with all fields encoding the same vector relation with the objects[29]. Identifying all cells using this mechanism is difficult in typical cue-rich environments, considering that cues other than objects might be encoded. Therefore, a simplified landscape is desirable for dissecting place field mechanisms. Ideally, landmarks should be sensed one at a time, and the animal's trajectory through the landmarks should be consistent over many trials. For this purpose, we used a treadmill apparatus, in which the only useful landmarks were small objects fixed on the belt, and in which mice ran with their head restrained[30]. We recorded in both hippocampal CA1 and CA3 regions using multi-site silicon probes, and we examined the impact of landmarks and landmark manipulations on the firing fields of pyramidal cells.

We observe two fundamentally distinct groups of cells in CA1. In one group, cells are akin to landmark-vector cells as they exhibit several fields with similar distance relationship to landmarks, and are referred to as LV cells for convenience. Cells in the other group are labelled context-modulated cells (or CM cells) since they exhibit single firing fields specific to a particular layout of objects on the belt. We show that LV cells are by an order of magnitude more frequent in CA1 than in CA3, and concentrate in the deep portion of CA1 pyramidal layer. In support to a larger involvement of sensory inputs compared with CM cells, LV cells are active across different environments and show instantaneous responses to object manipulation. We also show that LV cells discriminate landmarks based on their identity and that the probability for a landmark to be represented depends on its saliency. These findings demonstrate a functional organization of place field mechanisms, and bring new insights to the underlying mechanisms of landmark-vector representation.

## Results

**Context-modulated cells and landmark-vector cells.** To investigate the impact of various landmarks, we trained head-fixed mice to run for water rewards on a long treadmill belt (1.8–2.3 m) displaying a particular layout of landmarks (Fig. 1a). Importantly, the treadmill was not motorized, but consisted of a light velvet belt resting on two 3D printed wheels, which mice moved themselves at will[30]. The landmarks were fixed on the belt and were composed of vertically erected flexible objects or horizontally laid objects, lined along the edges of the belt, providing visual-tactile stimulation to both sides of the mice without interfering with their locomotion. We used four types of landmarks, of identical lengths (10 cm) but contrasting colours, textures and heights: an array of ~3 cm high glue spines, an array of horizontal shrink tubes, an array of pieces of Velcro and an array of vertical tubes. To detect possible cell activity associated with a given landmark, each landmark was fixed to at least two locations on the belt. After three weeks of training, we performed recordings from the pyramidal layers of the CA1 and CA3 hippocampal regions using either one or two 8-shank silicon probes (64 channels) (Fig. 1b, see 'Methods' section). The total number of trials (complete rotation of the belt) performed during the recording sessions varied from 47 to 291 (89.3 ± 21.2, mean ± s.e.m). We recorded a total of 2084 neurons (CA1, 1450; CA3, 636), during 36 recording sessions (CA1, 25; CA3, 11), in 23 different mice (CA1, 16; CA3, 7) following standard criteria for unit detection and clustering[31–33] (Supplementary Fig. 1). Consistent with a previous report[30], a fraction of the cells active in the treadmill exhibited stable firing fields in specific positions on the belt. Among those cells, we noticed two types of activity: cells that selectively discharged in one specific area of the belt (Fig. 1c,d), which we will refer to as CM cells; and, cells that exhibited firing fields tightly coupled to the landmarks on the belt, repeating in similar fashion at multiple landmark positions, in several cases regardless of landmark types (Fig. 1c,d, see 'Methods' section). Because of similarities with the LV cells[3,29] previously reported in 2D environments, we will refer to this second group as LV cells.

**Distinct anatomical organization of CM and LV cells.** We first compared the distributions of CM cells and LV cells across CA1 and CA3 regions. In contrast to CA1 cells, CA3 cells mostly exhibited single fields (CA1 $n = 299$, CA3 $n = 85$) and contained very few LV cells (CA1 $n = 209$, CA3 $n = 5$). The distributions were significantly different ($P = 0$, $\chi^2$ null hypothesis of independence, $\chi^2 = 119.7$, degrees of freedom: $k = 2$).

Within CA1, we examined the cell's locations along the radial axis of CA1 pyramidal layer, since distinct patterns of gene expression, connections and firing activity were reported in the superficial (CA1s, closer to S. Radiatum) and deep (CA1d, closer to S. Oriens) portions of the layer[23–28]. For this, we first estimated the position of each cell relative to the shank of the silicon probe, based on spike amplitude distribution across the recording sites (Fig. 2a–c; Supplementary Fig. 2, see 'Methods' section). Then, since each shank likely reached a different depth of the CA1 pyramidal layer, we estimated for each shank the position of

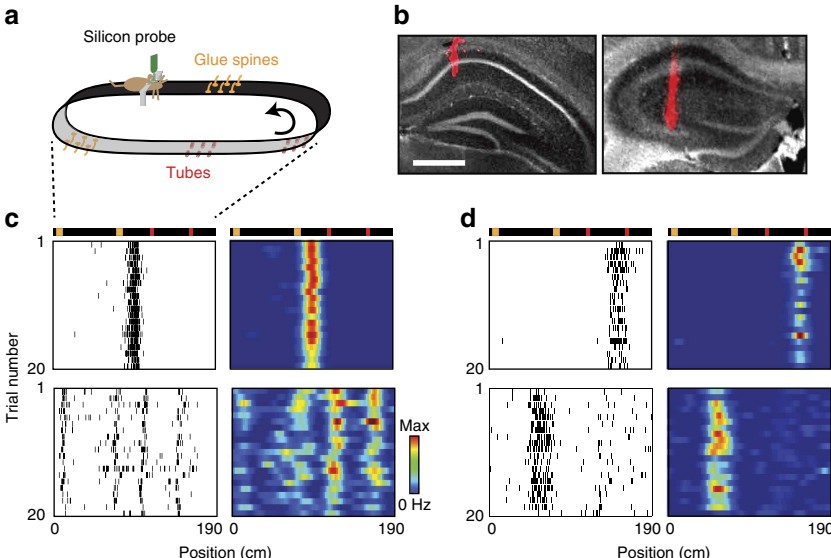

**Figure 1 | Context-modulated cells and LV cells in CA1 and CA3. (a)** Experimental setup for silicon probe recording in head-fixed mice during treadmill running. Visual and tactile landmarks of the belt are illustrated in different shape and colour. Landmarks were repeated in at least two different locations on the belt. **(b)** Superimposed DiI (red) and DAPI (grey) signals showing one shank of a silicon probe targeting distal CA1 (left) and CA3 (right). Calibration bar, 0.5 mm. **(c)** Example of CM cells and LV cells in CA1. Schematic representation of the belt (top), spike raster and firing rate as a function of belt position. Numbers on colour plots indicate peak firing rates. **(d)** Example of CM cells in CA3.

the recording site with maximum ripple power, and expressed cell depths in terms of distance from that position[24]. We observed that LV cells were concentrated in a deeper part of the layer than CM cells, as LV were located on average $4.4 \pm 2.8\,\mu m$ above ripple peak position while CM cells reside on average $-8.2 \pm 3.2\,\mu m$ below (Fig. 2d, LV cells: $n = 62$, CM cells: $n = 83$, $t_{143} = 2.7$, $P = 0.0077$, two-tailed unpaired $t$-test). To confirm these findings with an alternative method not involving the ripple peak estimation, we estimated the position of cell types relative to each other, by considering only shanks that contained cells of the two types. For each shank, we computed the difference in depth for all possible pairs between the two cell types. LV cells were systematically higher on the shanks than CM cells, by $20.1\,\mu m$ on average (Fig. 2e, $n = 538$ pairs, $t_{537} = -18.7$, $P < 0.0001$, one-tailed $t$-test), meaning that LV cells were located deeper in the pyramidal layer compared with CM cells, consistent with LV cells occupying CA1d and CM cells belonging mainly to CA1s.

We then examined the distribution along the proximo-distal axis (Fig. 2f, Supplementary Fig. 2, see 'Methods' section), since the relative proportion of LEC over MEC inputs is reported to increase toward the distal region[11,19]. Yet, both cell types could be found over the whole proximo-distal axis (LV distribution: $n = 123$, $P = 0.12$, CM distribution: $n = 89$, $P = 0.96$, Kolmogorov–Smirnov uniformity test) with no significant difference in cumulative distribution between the two cell types (Fig. 2f, $P = 0.24$, unpaired Kolmogorov–Smirnov test).

**Landmark specificity.** In previous studies on LV cells, the landmarks used were all different, and it was unclear if the landmarks encoded by a given LV cell were selected based on their physical identity, their saliency, or their location. We found that the identity of landmarks was encoded in a subset of LV cells, as their firing activity was stronger or exclusive to the positions of a particular landmark (Fig. 3a). To quantify this, we considered sessions ($n = 6$ sessions from four mice) in which the belt contained two landmarks of similar size (spines and vertical tubes). We first

identified for each cell the strongest firing field and called the landmark it encoded dominant landmark (versus secondary landmark for the other) (Fig. 3b). We defined an identity index as the difference, after normalization, in peak firing rates between dominant and secondary landmarks, considering only the smallest field of the dominant landmark and the largest field of the secondary landmark (Fig. 3b). An index value above zero indicates that all fields encoding the dominant landmark have higher peak rates than any of the fields encoding the secondary landmark. Large index values (close to 1) correspond to large rate differences between the two landmarks. We found that 49% (55/113 cells) of LV cells had identity indexes above zero, with 35 cells (63%) encoding the tubes and 20 cells (37%) encoding the spines. To test the significance, we compared the distribution of identity indexes with a shuffled distribution, obtained from a bootstrap procedure where the landmark identity of the fields for each cell was shuffled 10,000 times (Fig. 3b). A total of 21 cells (19%) had indexes exceeding the 95th percentile of the shuffled distribution (expected, 5.47 cells, $P = 10^{-7}$, Binomial test). Among these, 12 cells encoded the tubes while 9 cells encoded the spines, indicating that the underlying mechanism for specificity was the distinct identity of the landmarks and not simply a larger saliency of one of the landmarks.

Furthermore, we found that landmark saliency also played a key role. In another subset of recording sessions (CA1, $n = 8$ sessions from six mice) where the belt contained landmarks of diverse sizes (spines, Velcro, glue drops), we found that the spines, which likely provided the most intense visual and tactile stimulation because of their 3 cm height (compared with 5 mm at most for the other landmarks), were represented from 10 to 30 times more than other landmarks (Fig. 3c, (all) spine-versus-other pairs, $n = 8$, spine versus glue: $t_{14} = 3.48$, $P = 0.0037$, spine versus tube: $t_{14} = 3.46$, $P = 0.0038$, spine versus Velcro: $t_{14} = 3.36$, $P = 0.0047$, two-tailed unpaired $t$-tests).

**Field-to-landmark distance and field shape.** Fields encoding distances to landmarks should form a continuum to map the whole environment. We observed that in LV cells, the distances

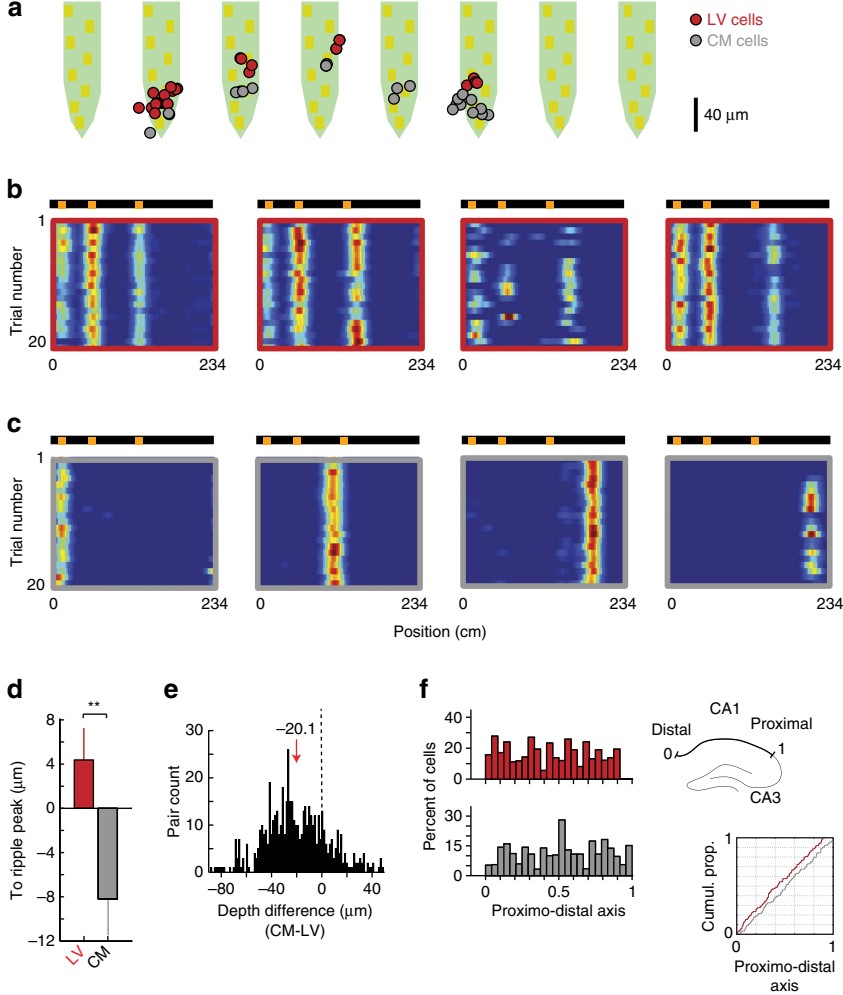

**Figure 2 | Cell repartition along CA1 radial and proximo-distal axes.** (**a**) Example of LV (red) and CM (grey) cells location along the eight shanks of a silicon probe with overlaid silicon probe geometry. Representative LV (**b**) and CM (**c**) cells recorded in **a**. (**d**) Average distance of LV (red) and CM (grey) cells from ripple peak positions, LV cells: $n = 62$, CM cells: $n = 83$, $P = 0.0077$, two-tailed unpaired $t$-test, \*\*$P < 0.01$. (**e**) Distribution of depth-differences between LV–CM pairs of neurons from the same shank. (**f**) Left, histogram showing the proportion of LV (red) and CM (grey) cells along the proximo-distal axis of CA1. Top, scheme showing the normalized disto-proximal position from 0 to 1. Right, corresponding cumulative distributions.

between fields and landmarks varied from one cell to another (Fig. 4a) in a seemingly continuous manner, but within an asymmetric distribution relative to the landmarks, mapping only positions where the mice could presumably see or touch the landmarks: while a substantial fraction of cells (48/209 cells, 23%) were anticipatory, that is, encoded positions up to 13 cm before the landmarks, the majority of the cells (161/209, 77%) encoded specific positions inside the landmarks, and virtually no cell encoded positions after the landmarks (Fig. 4b). Importantly, the field-to-landmark distances were preserved across all field repetitions in individual cells, as evidenced by a significant correlation between the different field-to-landmark distances ($n = 399$, $r = 0.56$, $P < 0.0001$, Pearson coefficient, Fig. 4c). Likewise, the field amplitude (peak rate) was maintained across field repetitions ($n = 399$, $r = 0.95$, $P < 0.0001$, Pearson coefficient, Fig. 4d).

We next compared the field dimensions of LV cells and CM cells. The average shape and amplitude of LV fields (10% edges width: $34.71 \pm 1.09$ cm, amplitude: $5.77 \pm 1.41$ Hz) was very similar to the average shape and amplitude of CM fields (10% edges width: $33.17 \pm 0.94$ cm, amplitude: $5.51 \pm 0.96$ Hz, Fig. 4e,f, LV: $n = 299$, CM: $n = 209$, $t_{506} = 0.14$, $P = 0.89$,

two-tailed unpaired $t$-test). Importantly, theta phase precession was present for both types of cells, with equivalent magnitudes and rates (Supplementary Fig. 3).

**Changing the belt**. Place fields are generally specific to the context, with small changes of contextual cues inducing rate remapping and larger changes producing global remapping. To test the context specificity of LV and CM cells, we performed consecutive recordings of the same cells in two different belts (belt A and belt B), which had distinct lengths and landscapes of objects (Fig. 5a).

First, we looked if cells could switch types between the two belts. For this, we compared for each cell the object scores in belt A and belt B. No CM cell was seen to convert into a LV cell from belt A to belt B, and conversely, no LV cell converted into a CM cell (Fig. 5b). Second, we asked how the firing rate activity was affected by the change of belt. LV cells firing activity was quite robust across the belts, with most LV cells showing firing fields in the two belts. This was despite the fact that the landmarks used in the two belts were different, implying that LV cells encoded distinct landmarks in belt A and belt B

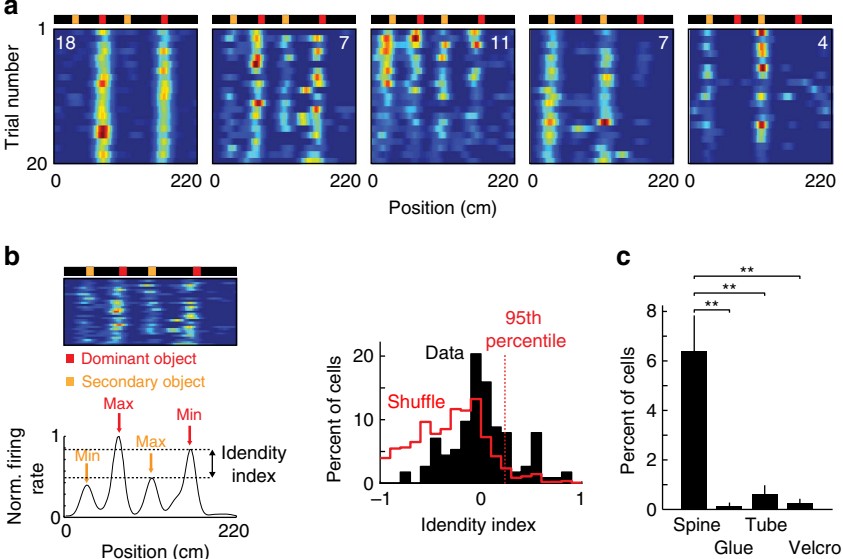

**Figure 3 | Representation of landmark identity by LV cells.** (**a**) Example of 5 LV cells recorded simultaneously, ordered from specific to spines (left), non-specific (middle) and specific to vertical tubes (right). (**b**) Left, each cell was normalized by the largest field and the landmark it encoded was designated as 'dominant'. The identity index was defined as the rate difference between the smallest field of the dominant landmark and the largest field of the secondary landmark. Right: Distribution of identity indexes for observed (black) and shuffled (red) data. (**c**) Repartition of LV cells by landmark type. Mean ± s.e.m., $n = 8$ sessions, **$P < 0.01$, two-tailed unpaired $t$-test. Exact $P$-values, $t$-statistic and degree of freedom reported in the text.

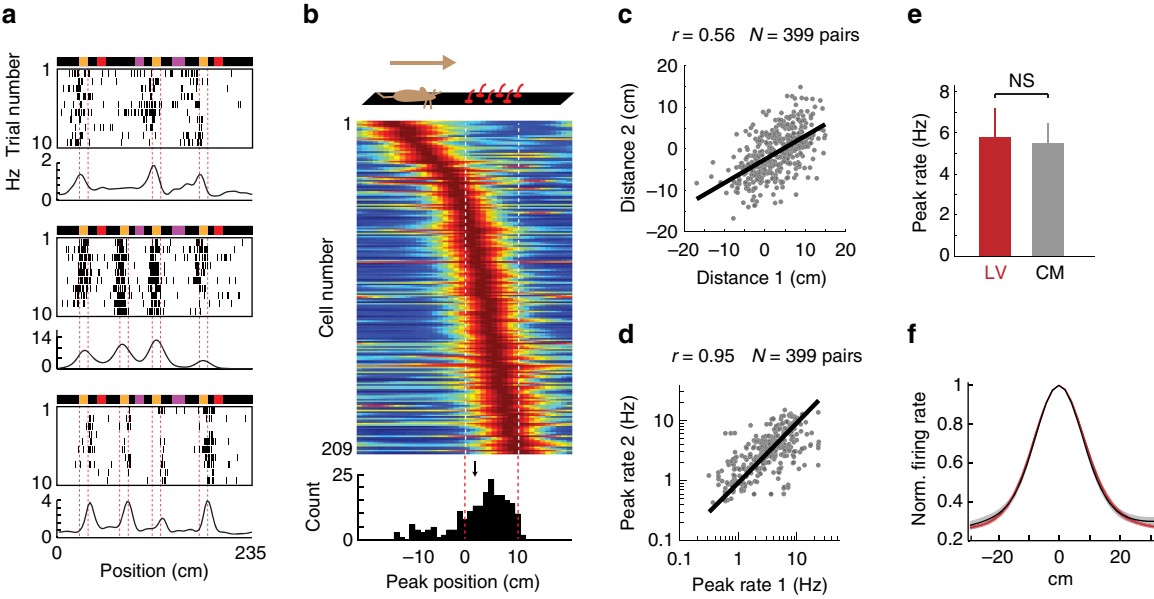

**Figure 4 | LV field characteristics.** (**a**) Example of cells with different field-to-landmark distances, that is, with fields encoding position at the beginning of the object (top), in the landmark (middle) and at the end of the landmark (bottom). (**b**) Distribution of field-to-landmark distances. Colour-coded, each row is the average of all firing fields of one neuron. Cells were ordered according to their field-to-landmark distance. Bottom, histogram of the distribution. The arrow indicates the mean. (**c**) Correlation of field-to-landmark distances in individual cells. Each point indicates the field-to-landmark distances of a pair of fields belonging to one cell. (**d**) Correlation of field peak rates in individual cells. Each point indicates the peak rates of a pair of fields belonging to one cell. (**e**) Peak rate, and (**f**) normalized fields average shape, for LV (red) and CM (grey) cells.

(Fig. 5a). In contrast, CM cells tend to be selective to one of the belts. Consistent with this, the fields' amplitudes were highly correlated between the two belts for LV cells ($n = 53$, A/A′: $r = 0.90$, $P < 0.0001$, Pearson coefficient. A/B: $r = 0.81$, $P < 0.0001$, Pearson coefficient) (Fig. 5c), but not for CM cells ($n = 46$, A/A′: $r = 0.76$, $P < 0.0001$, Pearson coefficient. A/B: $r = 0.26$, $P = 0.083$, Pearson coefficient) (Fig. 5d). To further quantify this, we estimated for each cell the rate overlap between the belts[34],

defined as the ratio of peak rates between belt A and belt B (belt A over belt B if belt B has the largest peak rate, and vice versa). The rate overlap of LV cells was significantly higher than for CM cells between belt A and B (Fig. 5e, LV: $n = 53$, CM: $n = 46$, $t_{97} = -4.11$, $P < 0.001$, two-tailed unpaired $t$-test). Finally, we examined if LV cells field-to-landmark distances were affected. Field-to-landmark distances tended to remain the same, showing a small but significant correlation between the

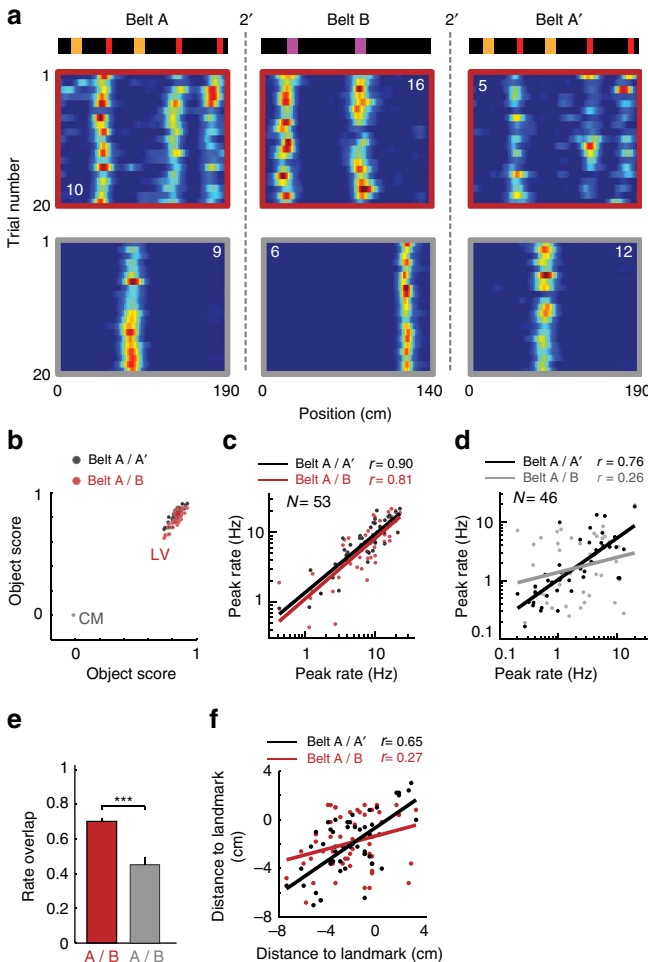

**Figure 5 | Distinct response of LV and CM cells to belt substitution.**
(**a**) Example of LV (red) and CM (grey) firing activity in two different belts.
(**b**) Object score of LV cells in belt A (x axis) versus belt B (red dots) and
A′ (second session of belt A, black dots). Notice that no LV cells became
CM cells (grey circled dots) and vice versa. (**c**) Peak firing rates of LV cells
in belt A (x axis) versus belt B (red) and A′ (black). (**d**) Same as **c** for
CM cells. (**e**) Rate overlap between LV (red) and CM (grey) cells in belt A
and B. LV, $n = 53$, CM, $n = 46$, $P < 0.001$. (**f**) Field-to-landmark distance of
LV cells in belt A (x axis) versus belt B (red) and A′ (black). r values,
Pearson correlation coefficient, ***$P < 0.001$.

two belts (A/A′: $r = 0.65$, $P < 0.0001$, Pearson coefficient.
A/B: $r = 0.27$, $P = 0.046$, Pearson coefficient) (Fig. 5f). This
was despite the fact that the landmarks were different, suggesting
that the mechanisms underlying landmark specificity and
field-to-landmark distances are independent.

**Instant dynamics of LV cells**. The mechanisms underlying
place field remapping have mostly been studied at low temporal
resolution, without taking into account the heterogeneous types
of place cells. To investigate these mechanisms, we either
removed a spine landmark, or added an extra one to the belt,
at a given point in the recording session.

We first examined the impact of these manipulations on
LV cells. LV fields tightly depended on the presence of the
landmark, as they disappeared instantly when the landmark was
removed (Fig. 6a, 3 sessions from 3 mice, $n = 11$ cells). The firing
rate measured in the landmark vicinity (by averaging the

firing rate in a 30 cm window around the landmark) reached
on average its asymptotic floor level the first trial the mice
experienced the landmark absence (Fig. 6b). Moreover, traces of
field activity could not be detected in individual cells after the
landmark removal, with the firing rate value in each cell reaching
the background level, defined as the mean firing rate in the
two 15 cm windows flanking the 30 cm window centred around
the landmark (Fig. 6c). Importantly, the fields in the remaining
locations of the landmark maintained the same firing rate
intensity throughout the session (Fig. 6a,b).

When an extra spine landmark was added to the belt, new
fields were created instantly in all LV cells (Fig. 6a–c, 4 sessions
from 3 mice, $n = 26$ cells), with the same field-to-landmark
distance and peak amplitude as pre-existing fields. The emergence
of the new fields was immediate, with the firing rate in the
landmark vicinity reaching on average its asymptotic value on
the first trial the mice experienced the added landmark (Fig. 6b).
At the level of individual cells, the field-to-landmark distance
relation was also apparent from the first trial (Fig. 6d), suggesting
altogether a pre-configuration of the underlying circuits. To test
further this idea, we examined the change in LV cells population
vector activity over time, by computing the population vector in
each trial, for positions within a 30 cm window around the added
landmarks, and then correlating this with a reference population
vector computed using late-session trials (trials 40 to 80,
see 'Methods' section). This analysis indicated an instant switch
of population activity to near stable state (Fig. 6e).

The landmarks involved so far were familiar to the mice due to
the 3 weeks training period. To see how novel landmarks were
represented, we added during the session a novel landmark
(vertical plastic tubes) that the mice had never encountered,
at two positions of the belt (Fig. 6a, 3 sessions, 3 animals). In
a fraction of cells (26/289 recorded cells, 9%), two firing fields
appeared instantly at the landmark locations. The emergence of
the fields was instantaneous, with the firing rate intensity
reaching on average asymptotic value on the first trial
(Fig. 6a,b). As in the familiar spine landmark experiments, the
field-to-landmark distance relation was apparent from the first
trial (Fig. 6d), suggesting that the mechanism underlying field-to-
landmark distances does not depend on landmark familiarity. At
the population level, the evolution of the population vector was
similar to the one for familiar landmark addition (Fig. 6e).

**Slower dynamics of CM cells**. We next investigated the remap-
ping dynamics of CA1 and CA3 CM cells subsequent to landmark
manipulation. Since these effects were similar for familiar and
novel landmarks, we pooled the data from both experiments.
While the addition of landmarks had no impact on a fraction of
CA1 ($n = 70$, 35.53%) and CA3 ($n = 24$, 37.84%) place cells,
they triggered field reconfiguration in a large number of cells
(CA1, $n = 127$, 64.47%; CA3, $n = 46$, 62.16%). In contrast to
LV cells, this remapping process was slow and involved distinct
dynamics, including 'switching' and 'drifting', as they were
characterized, respectively, by the gradual emergence of new place
fields in initially silent cells (CA1, $n = 80$; CA3, $n = 25$) (Fig. 7a;
Supplementary Fig. 4), and gradual drifts of pre-existing place
fields (CA1, $n = 47$; CA3, $n = 21$) (Fig. 7b, Supplementary Fig. 4).

The switching process (Fig. 7a) was neither immediate, nor
synchronous across the cell group, but instead was spread over
time, with some cells turning ON in the first trial following
landmark addition, and others several trials later (up to 68 trials
after addition) (Fig. 7c). The temporal rate of field creation
followed nevertheless an apparent exponential decay, with most
fields being created in the initial trials while gradually less
during subsequent trials. Similar trends were observed in CA3

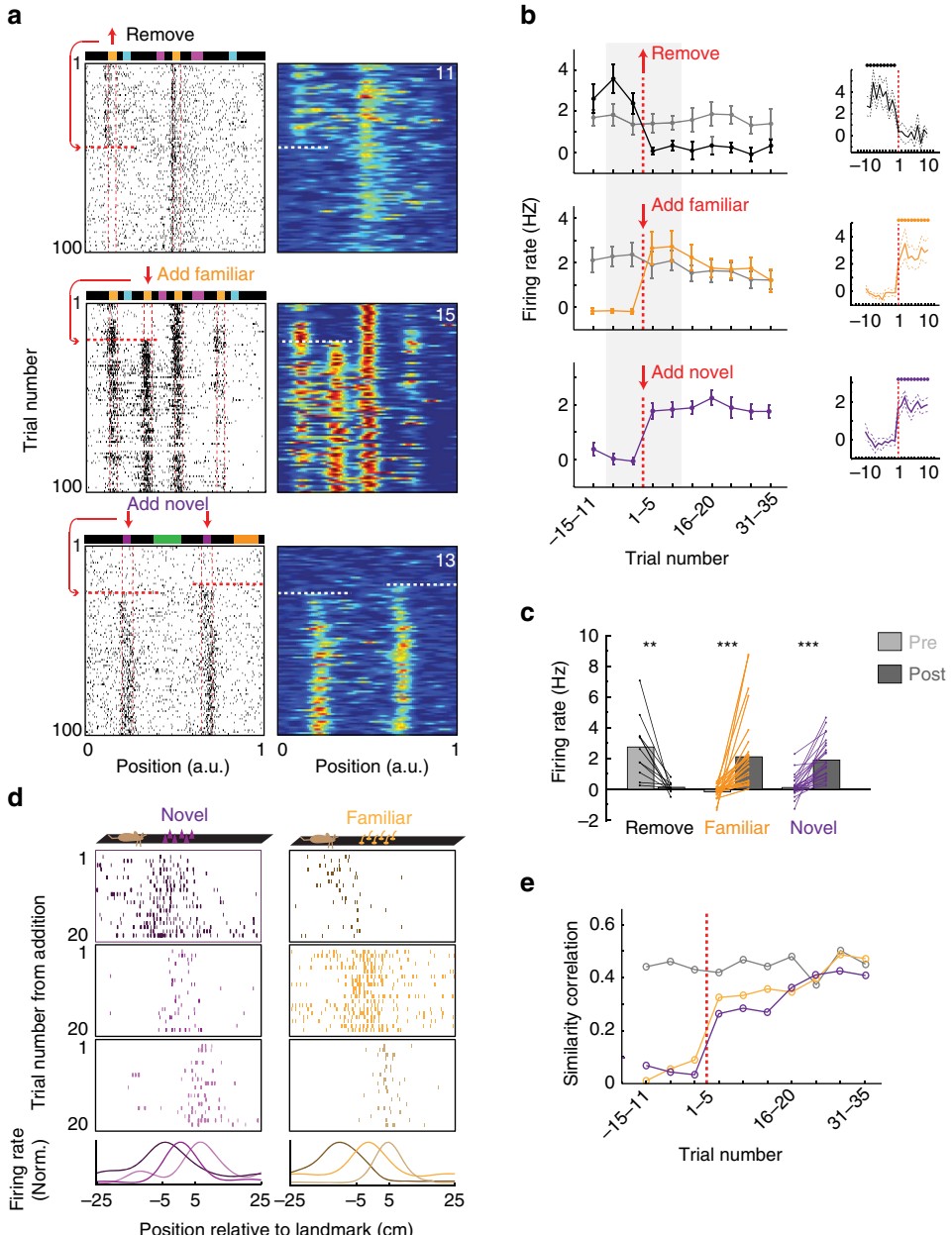

**Figure 6 | LV fields dynamics during landmark manipulations.** (**a**) Spike raster (left) and firing map (right) for three cell examples. Familiar landmark removal (top), addition (middle) and novel landmark addition (bottom). Red arrows and dashed lines indicate position and trial number for added/removed landmarks. (**b**) Mean firing rate at the position of the removed (black), added familiar (orange) and added novel (purple) landmarks. Grey, untouched spine landmarks. Background activity levels were subtracted (see 'Methods' section). Shaded box, single-trial precision. Dots at the top indicate trials with significant mean rate above 0. Remove: n = 11, add: n = 26, new: n = 26, significance level of 0.05, right-tailed t-tests. (**c**) Pre- and post-manipulation mean firing rates. Each cell is represented by linked dots. (**P < 0.01, ***P < 0.001, two-tailed paired t-tests). (**d**) Spike raster in landmark-centred window for three cell examples for the first 20 trials after novel (left) and familiar (right) landmark addition. Note that field-to-landmark distances are apparent from the first trial. (**e**) Evolution of population activity.

(Supplementary Fig. 4). Importantly, the newly created place fields were gathered in the immediate vicinity of the added landmarks (Fig. 7d).

On the other hand, pre-existing place fields occasionally started drifting after the addition of a landmark (Fig. 7b; Supplementary Fig. 4). To quantify this effect, we tracked the centre of mass of each place field over the trials (Fig. 7e; Supplementary Fig. 4, see 'Methods' section), and define as drifting the ones that drifted on average by more than 0.1 cm per trial in one direction. The drifts could range from 6.6 to 102 cm (46.23 ± 4.34 cm,

mean ± s.e.m) and last from between 22 and 189 trials (99.29 ± 7.32 trials, mean ± s.e.m) with rates ranging from −0.46 to 0.28 cm per trial (−0.19 ± 0.02 cm/trial, mean ± s.e.m). Reminiscent of previous reports of backward shifts in place fields[35,36], drifts evolved mainly in the direction opposite to motion (41 cells backward versus 6 cells forward) (Fig. 7f; Supplementary Fig. 4).

Finally, switching cells and drifting cells were found at distinct depths along the radial axis of CA1 pyramidal layer, occupying respectively CA1d and CA1s (Supplementary Fig. 5).

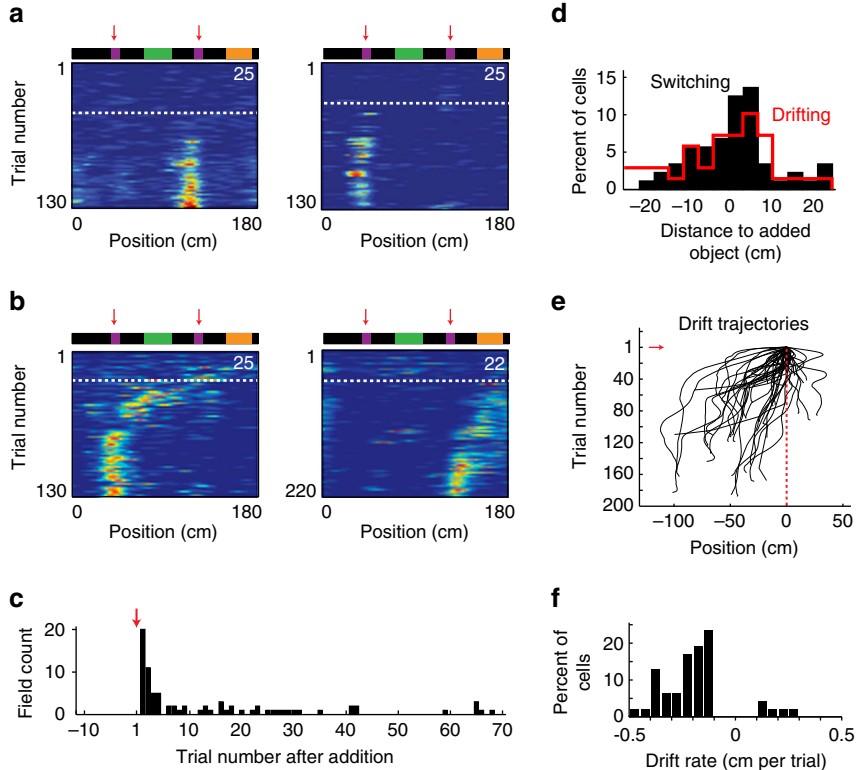

**Figure 7 | CM fields dynamics following landmark manipulations.** (**a**) Example of 'switching' and (**b**) 'drifting' cells in CA1. (**c**) Field emergence of switching cells as a function of trials. (**d**) Distribution of field positions relative to the added landmark for switching (black) and drifting (red) cells. The landmark is 10 cm long and centred around zero. (**e**) Trajectories of drifting place fields along the trials. Field drift starting positions are aligned on zero. (**f**) Distribution of drift rates.

## Discussion

Our findings support an anatomical segregation of LV cells to the deeper portion of the CA1 pyramidal layer. *In vivo* physiological differences between CA1d and CA1s were previously reported[24,27] but focused on spike phase relationships with theta and gamma rhythms, entrainment by slow wave sleep rhythms, burst activity, number of place fields and ripple activity. To the best of our knowledge, this is the first time a specific place field mechanism is matched to a particular region of CA1. In terms of afferents, CA1d receives most inputs from the region CA2 (ref. 22) and maybe from the entorhinal cortex[24]. On the other hand, CA1s pyramidal cells are relatively more controlled by CA3 inputs, as they are excited by CA3 stimulations and sharp-wave ripple events, while CA1d cells exhibits mainly inhibitory responses[27] (presumably through concerted CA3-to-CA2 and CA1s-to-CA1d feed-forward inhibitions[22,28]). Hence, CA1d and CA1s might belong to two distinct streams of information, CA2-CA1d and CA3-CA1s, respectively. In this respect, our finding that LV cells are located in CA1d matches recent evidence of strong object influence on CA2 activity[37,38], and suggest CA2-CA1d as a more sensory stream. In contrast, the CA3-CA1s stream is likely involving more memory-related mechanisms because of CA3 extensive recurrent collaterals[39,40]. Accordingly, cells in both CA3 and CA1s showed single firing fields and slower dynamics. Single fields could not arise from simple visual-tactile sensory mechanisms since every landmark was repeated at least twice on the belt. It is also unlikely that they arise from odours on the belt, since in absence of visual-tactile cues, very few cells have place fields and none retain their position when the reward location is moved[41] (fields correlated either with travel distance[42] or reward distance[41]). Instead, we suggest

that single firing fields arise within CA3 from the encoding of conjunctions between sensory, path-integration and local recurrent inputs, and then are relayed to CA1s.

Landmark-vector cells were previously reported in a study from Deshmuck *et al.*[29], but in smaller proportion than the LV cells described here. This has several possible explanations. First, it is possible that a fraction of LV cells failed to be identified in that study. Indeed, all objects used were different, and considering our finding that LV cells encode landmarks identity, it is possible that some LV cells exhibited only single fields and were therefore missed. In addition, it is possible that some LV cells were encoding environmental cues other than the objects, such as maze corners. Second, our landmarks were designed to provide overwhelming whiskers/body stimulation, and the mice had to run through the landmarks. Hence, they likely generated a more intense sensorial stimulation than the objects used in the study of Deshmuck *et al.* This should be an important factor considering our finding that landmark saliency is critical for LV cell representation. Third, it is possible that the number of LV cells is inflated in the treadmill because the sensory information is oversimplified. Indeed, cells probably use a range of other sensory information in two-dimensional arenas, such as head direction and distal cues, which might usually compete or integrate with local landmarks. This might not necessarily be an artifact of the treadmill, but a difference between one and two-dimensional environments. Indeed, it is worth noting that in a study[43] where local cues were laid on a linear track, place cells similar to LV cells were reported in significantly large numbers. These cells had bidirectional place fields that encoded in each direction an equidistant position ahead of a landmark, and were suggested to reflect view-invariant object information.

Also, in a study where rats had to run through a one-dimensional zigzag pattern, a large fraction of cells in CA1 showed a repetition of place fields at equivalent positions of the zigzag trajectory[44]. Similar to here, such cells were much less frequent in CA3. Last, it is possible that the quantitative discrepancy between the two studies reflects differences between mouse and rat species, since rats were used in the study of Deshmuck et al. Compared with rats, mice might use more the local cues over the distal cues[45], resulting in a larger number of LV cells in mice.

Our results provide new insights on the mechanisms underlying LV cells. LV firing fields were closely associated with sensory mechanisms, as they showed repetitions and instant dynamics, but encoded both spatial (landmark distance) and non-spatial information (landmark identity and saliency). While non-spatial object information is believed to reach the hippocampus via LEC[14,15], spatial information might come from MEC inputs. Hence, a possible scenario is that LV cell activity emerges from an interaction of LEC and MEC inputs, contributing respectively the landmark specificity and landmark distance aspects. For instance, considering that grid cells reset and activate at similar distances in the repeated alleys of a hairpin maze[46], and that border cells activate near boundaries[18], it is possible that some grid cells and border cells encode particular positions near the landmarks, supplying the hippocampus with discrete spatial inputs, which sum with the object specific (but less spatially tuned[14,15]) inputs from LEC.

In common with LV cells, switching cells were found in the deep CA1 pyramidal layer (CA1d), and developed new firing fields near objects added to the belt. This process, however, was more gradual, with new fields emerging after tens of trials, suggesting a progressive network buildup involving synaptic plasticity mechanisms[47]. It is tempting to see LV cells as part of a continuum with switching cells, expressing the largest prevalence of landmark-related sensory information over contextual information, and being followed by early and then late switching cells.

More superficially located in the CA1 pyramidal layer, drifting cells were likely the least controlled by landmarks, as drift of fields largely suggests a dissociation between field mechanisms and landmarks inputs. As a mechanism, drifts are reminiscent of backward shifts in freely moving rat experiments, during initial maze exploration[35] or after cue rotation[36], with the difference that drifts could span up to 100 cm compared with the 2–10 cm of backward shifts. It has been proposed that backward shifts emerge from the combination of spike theta phase precession and the asymmetric nature of spike time-dependent plasticity (STDP)[48,49]. Field drifts in the treadmill might also arise from such mechanisms, and be exclusive to CA1s for a number of reasons including differences in inputs[22] and local circuits[28], and the fact that CA1s pyramidal cells contain calbindin and zinc, two molecules involved in synaptic plasticity[23].

Our findings suggest a functional division between CA1 deep and superficial layers. While LV cells in the deeper layer supply sensory mediated representation of self-position and object locations, cells with looser ties to landmarks tackle spatial representation on a more global level, using both sensory and memory information, and may also be more flexible for integrating non-spatial factors such as reward, goal and time[7,9,10]. The coexistence of these distinct place field mechanisms suggests that diverse types of spatial associations, involving distinct levels of specificity, precision and portability across environments, might occur in parallel. The fact that CA1d generates most CA1 projections to brain regions involved in goal oriented behaviours (ventral striatum/nucleus accumbens, septal area and orbitofrontal cortex)[23] might underlie a behavioural benefit for reward prediction mechanisms to be linked with discrete cues and

transferable across environments, while CA1s predominant feedback projections to the medial temporal cortex[23] might contribute the contextual information to episodic memory processes in this region. Future experiments using selective inactivation of deep and superficial CA1 cells should help reveal their relative contribution to memory.

## Methods

**Animals.** All experiments were conducted in accordance with institutional regulations (Institutional Animal Care and Use Committee of the Korea Institute of Science and Technology), and conformed to the Guide for the Care and Use of Laboratory Animals (NRC 2011). Overall, 23 male C57BL/6 mice between 6 and 7 weeks were used. The mice were housed 2 to 3 per cage, in a vivarium with 12 h light per dark cycles. Training and recording sessions described next occurred during the light cycles.

**Preparation for head fixation.** During a first surgery under isoflurane anaesthesia (supplemented by subcutaneous injections of buprenorphine $0.1\,\mathrm{mg\,kg^{-1}}$, and followed by daily subcutaneous injection of ketaprofen $5\,\mathrm{mg\,kg^{-1}}$ for 2 days), two small watch-screws were driven into the bone above the cerebellum to serve as reference and ground electrodes for the recordings. A 3D printed plastic head-plate with a window opening in the centre was cemented to the skull with dental acrylic. The head-plate was designed to be conveniently fixed (and unfixed) to a holding plate using two screws.

**Behavioural training.** After a post-surgery recovery period of 7 days, the mice were water restricted to 1 ml of water per day, and trained for 3 to 4 weeks (1-h session per day) to run on the treadmill with their head fixed. The treadmill was not motorized, but consisted of a light velvet belt laying on two 3D printed wheels, which mice moved themselves at will[30]. Sucrose-in-water (10%) rewards were delivered every trial at the same position of the belt via a lick port. The lick port was equipped with a light-emitting diode and photo-sensor couple that enabled detection of individual licks. Belts of different lengths (ranging from 169 to 234 cm) and displaying different number of cues were used depending on the experiments. After behavioural learning reached an asymptote, the animals completed 100 to 150 trials in the first 45 min of the sessions. The quantity of sucrose-in-water consumed in the treadmill was measured after each session, and additional water was provided such that the mice drank a total amount of $1\,\mathrm{ml\,day^{-1}}$.

**Recording procedures.** We performed both acute and chronic recordings (acute, 9 mice, 15 sessions; chronic, 14 mice, 21 sessions). While acute experiments allowed the usage of higher channel count silicon probes ($2 \times 64$ channels probes), chronic experiments were necessary, for instance, to record the same cells in different belts. Since similar results were obtained with both approaches, the two data sets were pooled.

For acute recordings, on recording days, the mice were initially anaesthhetized with isoflurane and installed with their heads fixed on the treadmill. Following a subcutaneous injection of buprenorphine ($0.1\,\mathrm{mg\,kg^{-1}}$), a craniotomy of $\sim 1\,\mathrm{mm}^2$ was performed using a stereotaxic manipulator on one of the hemisphere at a location centred 2.2 mm posterior to bregma and 1.5 mm lateral to the midline, and the dura was removed (on the subsequent day, the craniotomy was done on the other hemisphere). The backside of the silicon probes shanks were coated with a cell labelling red-fluorescent dye (DiI, Life technologies) using the tip of a foam swab. The silicon probes were then fixed to micro-manipulators and lowered into the brain at a speed of $\sim 50\,\mu\mathrm{m\,min^{-1}}$. The hole was then sealed with liquid agar (1.5%) applied at near body temperature. Aluminum foil was folded around the entire probe assembly, to serve as a Faraday cage. After the silicon probes reached the target area, the anaesthesia was removed. Mice typically recovered from anaesthesia after 30-45 min and then spontaneously started running in the treadmill for sucrose-in-water rewards. Recording sessions typically lasted for 70 min, during which the animal's behaviour alternated between periods of running and immobility. After each recording session, the probe was removed and the hole was filled with a mixture of bone wax and mineral oil, and covered with silicon sealant (WPI, Kwik-sil). Individual mouse was recorded for a maximum of three sessions (one session per day).

When the mice woke up in the treadmill after the craniotomy/probe insertion procedures, no signs of distress were visible from either behaviour or local field potential signals. Behavioural signs of distress, such as mice struggling and grabbing the side posts, were visible only when mice initially experienced head-restriction during training, and were completely absent at any stage of the recording sessions. Typically, after the anaesthesia was turned off, local field potential progressively started showing quiet sleep associated ripple oscillations. The first detectable movements were usually occasional lickings, happening during a period of somnolence/ripple activity. This period was useful for shank stabilization and for confirming CA1 location by ripple activity. Mice typically started performing the task as soon as they began to move the belt.

For chronic recordings, a similar craniotomy was performed under isoflurane anaesthesia. A silicon probe was mounted on a custom-made micro-drive, and inserted one millimetre above the pyramidal layer. The micro-drive was cemented to the skull and head-plate. Bone wax and mineral oil mixture was used to cover the craniotomy. Then, the silicon probe was slowly lowered to the pyramidal layer using the micro-drive. A plastic cap was used to protect the micro-drive/silicon probe assembly. Recordings were performed starting the next day, one session per day, and for up to three sessions.

**Anatomy.** On the last day of recording, the animals were anaesthhetized at the end of the recording and perfused transcardially with 4% paraformaldehyde in phosphate buffer. The brain was removed and kept overnight in 4% paraformaldehyde solution.Overall, 100 µm thick coronal sections were cut using a vibratome and mounted on slides using Vectashield mounting medium with dapi. Images of dapi and DiI fluorescence were acquired separately with a Nikon FN1 microscope equipped for fluorescence imaging.

**Behaviour control and data acquisition.** The forward and backward movement increments of the treadmill were monitored using two pairs of LED and photosensors that read patterns on a disc coupled to the treadmill wheel, while the zero position was implemented by a LED and photo-sensor couple detecting a small hole on the belt. From these signals, the mouse position was implemented in real time by an Arduino board (Arduino Uno, arduino.cc), which also controlled the valves for the reward delivery. Position, time and reward information from the Arduino board was sent via USB serial communication to a computer and recorded with custom-made LabView (National Instruments) programs.

For acute recordings, neurophysiological signals were acquired continuously at 24,414 Hz on a 128-channels recording system (Tucker-Davis Technologies, PZ2-128 preamplifier, RZ2 bioamp processor). For chronic recordings, neurophysiological signals were acquired continuously at 30,000 Hz on a 250-channels recording system (Intan Technologies, RHD2132 amplifier board with RHD2000 USB Interface Board and custom-made LabView interface).

The wideband signals were digitally high-pass filtered (0.8–5 kHz) offline for spike detection or low-pass filtered (0–500 Hz) and down sampled to 1,000 Hz for local field potentials. Spike sorting was performed semi-automatically, using KlustaKwik (klustakwik.sourceforge.net)[32], followed by manual adjustment of the clusters with Klusters[33]. Further data analysis was done in Matlab.

**Implementation of single neuron firing rate vector.** The length of the belt was divided into 100 pixels. To generate a vector of firing rates, the number of spikes discharged in each pixel was divided by the time the animal spent in the pixel. The firing rate vector was smoothed by convolving a Gaussian function (15 cm half-height width).

**Detection of place fields.** To detect place fields, we estimated the significance of positive humps in firing rate by shuffling spike times. For each shuffle, the spike train was split in two at a randomly chosen time $t$, and the two parts were 'rotated' by shifting them by $+t$ and $-t$, respectively. The goal was to mix the temporal relation between spikes and behaviour, without affecting the temporal structure of the spikes. We computed the cells firing rate vectors for 1,000 shuffles. The $P$-value of each pixel was given by the percentage of shuffles having a firing rate value higher than the original data. Place fields were defined as firing rate humps that contained at least five consecutive pixels with $P$-values lower than 0.01.

**Detection of LV cells.** To be classified as a LV cell, a cell should first have a number of detected place fields greater than 1. We then defined a landmark score ranging from 0 to 1 as the maximum of the cross-correlogram between the firing rate vector of the cell and a 'belt template'. The belt template is an array of zeros and ones matching the position of the landmarks on the belt (1 inside the landmarks, 0 otherwise). To detect LV cells, landmark scores were recalculated for cells' spikes shuffling procedure similar as in the method for place field detection. Cells with landmark score exceeding the 95th percentile of the shuffle distribution were defined as LV cells.

**Estimation of LV firing rate changes and background level.** Landmark manipulation might induce firing rate changes but also field shifts and broadening. To avoid a contamination of the measure of firing rate by the latter, we looked at the evolution of average firing rate considering all pixels within a 30 cm window centred on the position of the added or removed landmark.

Many cells showed non-zero background firing activity. To disambiguate between background activity and firing field activity, we subtracted the background activity, which was defined as the average firing rate in two 15 cm windows flanking a 30 cm window centred on the landmark.

**Drift of place fields.** The drift trajectory of place cells was estimated by computing the position of the field centre of mass after each trial. Neurons exhibiting a drift rate higher than 0.1 cm per trial constituted the set of drifting cells.

**3D reconstruction.** Digital pictures of coronal slices DAPI and shanks DiI signals were loaded into Matlab. The contour of hippocampus CA and the DiI signal of the silicon probe shanks were detected. The entire hippocampus CA region and shanks were reconstructed in 3D, and visualized with different rotations using custom-made Matlab routines. Shank positions along CA1 medio-lateral axis were estimated as the normalized distance, following CA1 curvature, from the border of subiculum, where the borders of subiculum and CA2 were respectively position 0 and 1, and were defined according to the Allen Mouse Brain Atlas (see Supplementary Fig. 2)[11].

**Estimation of cell position relative to the shank.** To estimate the position of a cell relative to the recording sites of a shank, we assumed that the amplitude of spike signals attenuate as $1/d^2$ (see note below), where d is the distance of the site to the cell soma, such that the amplitude measured at a given site is:

$$a_i = A/d_i^2$$

with $A$ the spike amplitude exactly at the cell position.

For the several recording sites of one shank, this means that:

$$A = a_1*d_1^2 = a_2*d_2^2 = a_3*d_3^2 = a_4*d_4^2 = a_5*d_5^2.$$

Therefore, to estimate the position of a cell, we simply search for the position where these conditions were fulfilled. For this, the volume around each shank was divided in 1 µm³ pixels, and for each pixel we computed the Euclidean distances to each recording site. Then we defined a value S such that:

$$S = \sum_{ij} \left| a_i*d_i^2 - a_j*d_j^2 \right|$$

where i and j varied to generate all possible combinations of sites

The pixel with the smallest value of $S$ was defined as the cell position.

Note: Electric potential of dipoles attenuate as $1/d^2$ while as $1/d$ for monopoles. We tested the method using either form and found the resulting cell positions to be very similar.

**Statistical analysis.** All statistical analyses were performed in Matlab (MathWorks). Number of animals and number of recorded cells were similar to those generally employed. For each distribution, a Kolmogorov–Smirnov test was used to test the null hypothesis that the sample distribution was derived from a standard normal distribution. If normality was uncertain, we used non-parametric tests as stated in the main text or figures. Otherwise, Student $t$-tests were used to test the sample mean. Correlations were computed using Pearson's correlation coefficient.

**Data availability.** The data that were collected for this study are available upon reasonable request.

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

## Acknowledgements

We thank Kamran Diba, Larry Cohen and Bradley Baker for helpful comments on the manuscript. This work was supported by the Korea Institute of Science and Technology Institutional Program (Project Nos. 2E26190 and 2E26170) and the Human Frontier Science Program (RGY0089/2012) to S.R., and by the Brain Research Program through NRF funded by the Ministry of Science, ICT & Future Planning by the Korea Government (NRF-2015M3C7A1031395) to J.-S.C.

## Author contributions

T.G. and S.R. designed the experiments and analyses. T.G. and M.F. performed the experiments. T.G. analyzed the data with input from S.R. T.G. and S.R. wrote the manuscript with input from M.F. and J.-S.C.

## Additional information

**Competing financial interests:** The authors declare no competing financial interests.

