## [Peer Review File · Nature Communications]

Reviewers' comments:

Reviewer #1 (Remarks to the Author):

In this study, headfixed mice ran on a treadmill on which different surfaces were affixed to act as landmarks, while place cells from CA3 and CA1 were recorded via either acutely or chronically recorded silicon probes. The main findings are as follows:

- (1) One group of place cells (LVs) had multiple fields aligned or near the landmarks, while another group (SPFs) had single fields
- (2) The relationship of LV cells to landmarks was asymmetric, with cells having fields before the landmark or at it, but rarely beyond it
- (3) The LV cells tended to be found higher up the silicon shank (i.e. deeper in the CA1 cell layer, closer to stratum oriens). They were also relatively more numerous in CA1 than CA3.
- (4) When the treadmill belt was swapped for a different one, SPF cells remapped, again to a single field, while LV cells continued to fire near landmarks. The SPF changes were very slow while the LV response to the new landmarks on the new belt was essentially immediate.

This is an interesting study which offers to shed light on the way in which incoming inputs to the hippocampal system are organized, and points to a potential organization along the radial as well proximo-distal axes of the hippocampal cell layers. It is very well written (apart from numerous small grammatical errors) and clearly explained. However there are some issues to do with the conduct and interpretation of the experiments that are of concern.

To begin with, I have some ethical concerns. The procedure involved recording from animals that awoke from anesthetic to discover their heads fixed in a clamp and holes drilled in their skulls. A human who discovered themselves in this predicament would be frightened and in pain, but no mention is made of analgesia or any attempt to mitigate the trauma of this experience, e.g. by making a craniotomy ahead of time and allowing recovery from the painful part of the surgery first. We need to see a detailed justification of this procedure by the authors, including a weblink to the ethical guidelines under which they performed their procedures, a detailing of the considerations they gave to minimizing suffering, and the reasons why they did not take all possible steps to mitigate the experience for the animals.

In terms of the data, I think the characterization of two different types of place cell responding is interesting, but there is a question-mark over the novelty of this, given the previous report (which they cite) of "landmark vector cells" by Deshmukh and Knierim (2012). A simple way to characterize the findings might be that LV cells are responsive to landmarks in a mostly non-discriminating way (although the rate remapping illustrated by the "identity score" suggests a weak discrimination), while SPV cells are responsive perhaps to olfaction, and thus only fire in one part of the track (otherwise, if olfaction did not play a part, how would the cell know where to fire?). This relatively slight difference could explain why the LV cells responded so quickly to landmarks on the new belt (they were already responding to landmarks, and were not really discriminating them). The slow response of the SPV cells might be because the animal had to learn to detect the odors on the new belt

- that is, it is a product of perceptual learning/attention. Where in the circuit this learning was taking place is an interesting question.

The observation that the different responses were anatomically segregated along the radial axis of CA1 is also interesting, as is the way in which the cells responded to the different types of cues, and the fact that cells did not change their response patterns when the cues changed (with the second belt). However I would first like to see some more robust statistics to support the claim of anatomical segregation. The procedure of calculating pairwise distances between cells is flawed because it multiplies the data without justification - the correct comparison, to claim that the LV cells were deeper than the SPF cells, would be to calculate the depth of the two groups and then compare with a straightforward unpaired two-tailed (not one-tailed) t-test.

The spatial relationship of the LV fields to the landmarks raises some interesting questions about the mechanism and/or function.

Other comments

P4 What does "quasi absent" mean? Also, "robust across distinct belt environments" is not understandable at this point.

P5 The description of landmarks at this stage should make clear that the landmarks were on the belt and not fixed in the room (it took me a while to appreciate this point). A little later in that paragraph, it should be explained that the LV cells usually responded to all the landmarks regardless of type (though there was, as they show, some rate remapping).

What was the reason for having three different lengths of belt?

Was the distribution of LV and SPF cells different between CA3 and CA1 (by chi-square)?

P6 The "identity index" was unintuitive to me. A more natural (to my way of thinking) measure would have been the ratio of the peak of the best landmark to the peak of the secondary one; or a discrimination index (difference in the peaks divided by the sum). What was the reasoning for the measure used?

Reviewer #2 (Remarks to the Author):

It is well understood that place cell activity in CA1 is heterogeneous with different pyramidal neurons changing their firing patterns in response to different aspects of experience. This heterogeneity is strikingly different from CA3, in which neurons respond similarly to changes in experience. The varied responses between CA1 neurons have been attributed to the varied inputs from different anatomical regions, and the nature of a feedforward circuit. Studies have shown that such differences may be associated with the position of the cell along the proximal-distal axis of CA1 in which inputs are either more strongly driven by the

lateral or medial entorhinal cortex. The manuscript from Geiller et al. have expanded the dynamics of the heterogeneity of the CA1 network by showing that cells are also functionally clustered across the deep and superficial regions of CA1. Specifically, they recorded CA1 units from mice as they ran on a treadmill with many salient objects. The deep and superficial layers of CA1 can be clearly distinguished by the use of silicon arrays, with a precision that would not have been possible using traditional tetrode recordings. They find four main categories of cellular activity during behavior: 1) cells that fire specifically for a distinct landmark, 2) cells that fire for a position on the track, 3) cells that gradually drift during experience, and 4) cells that remap immediately and reliably to new objects. After reconstruction of recording sites they find that the anatomical location of the cells that show these 4 responses cluster across the radial axis with deep layers containing more landmark cells and superficial layers more classic spatial place fields. Across varied remapping conditions the identity of a cell did not switch between these categories.

The finding of this anatomical segregation is novel but the weakness of this work is that the origins or mechanisms are not explored or tested. This is where my enthusiasm for the findings was decreased. The authors favor the interpretation that different anatomical projections to CA1 that each represent different types of information underlie their findings. However, they do not test the influence of different inputs on CA1 activity by incorporating inactivations, lesions, or manipulations in mutant mice where individual inputs can be manipulated. Even without these additional experiments the manuscript would be stronger with an improved discussion of the anatomy and a presentation for a working hypothesis on how the varied inputs could result in their findings. Currently the authors remain non-committed suggesting a number of potential contributions or theories that are not well discussed.

1) The authors argue that the LEC and MEC inputs could potentially support the LV and SPF cells differentially. How do the authors then explain their findings in Figure 2 in which the proportion of LV and SPF cells does not change across the proximal-distal axis of CA1 where LEC and MEC inputs are known to be differentially segregated?

2) From the 3D anatomy reconstruction in Supp Figure 2 it appears that the majority of CA1 recordings are located in the distal region of CA1 closer to subiculum and few red lines appear to be located in the distal areas closer to CA2. How representative is the example shown. There should be a summary for the distal and proximal axis across animals in order to see the extent to which the total region was sampled across animals.

3) The authors need to show to what extent the proportion of cells they report across experiments is driven by individual mice or by anatomical segregation. For instance can the proportion of drifting vs. switching cells be explained by individual mice in which one mouse had more drifting cells and another more switching cells. For all experiments numbers need to be reported or data shown that verify that the phenomenon they describe and the proportions of cells they report hold within individual mice.

4) Can the authors clarify in the methods whether the speed of the track is controlled by the experimenter and is continuous at a set speed, or whether the animal can modulate this by

their own behavior.

5) the authors should discuss their findings and views of the circuits with the new data that has been recently reported describing activity in CA1d and CA1v during SWR activity (de la Prida).

Point-by-point response

Reviewer #1 (Remarks to the Author):

This is an interesting study which offers to shed light on the way in which incoming inputs to the hippocampal system are organized, and points to a potential organization along the radial as well proximo-distal axes of the hippocampal cell layers. It is very well written (apart from numerous small grammatical errors) and clearly explained. However there are some issues to do with the conduct and interpretation of the experiments that are of concern.

Point #1

To begin with, I have some ethical concerns. The procedure involved recording from animals that awoke from anesthetic to discover their heads fixed in a clamp and holes drilled in their skulls. A human who discovered themselves in this predicament would be frightened and in pain, but no mention is made of analgesia or any attempt to mitigate the trauma of this experience, e.g. by making a craniotomy ahead of time and allowing recovery from the painful part of the surgery first. We need to see a detailed justification of this procedure by the authors, including a weblink to the ethical guidelines under which they performed their procedures, a detailing of the considerations they gave to minimizing suffering, and the reasons why they did not take all possible steps to mitigate the experience for the animals.

The reviewer raises here important ethical concerns about potential pain and distress associated with the experiments in the treadmill.

First the reviewer is correct that we haven't provided enough details regarding the surgery procedures and post-surgery care, and we have revised our manuscript accordingly. We have strictly followed the *guide for the care and use of laboratory animal* (NRC 2011) and standard procedures approved by the IACUC of KIST. Surgeries were performed under isoflurane anesthesia and were preceded by subcutaneous injections of buprenorphine (0.1mg/kg). Subcutaneous injections of ketapofen (5mg/kg) were performed daily for 2 days after the surgeries. These steps are now specified in the methods.

Second, we carefully monitored the post-surgery recovery of animals. Regarding potential mice distress from waking-up head-fixed in the treadmill after the craniotomy/probe insertion:

- 1) It is important to point out that mice are apparently very comfortable being head-fixed on the treadmill. The treadmill is not motorized, the mice move themselves the belt whenever they want. The mice are trained for 3 weeks prior to the recordings. Signs of distress caused by the head-fixation, such as their legs grabbing the side posts, are seen on the first 2 days of training, completely disappear within a week, and were totally absent during the recording sessions. Mice apparently enjoy running on the treadmill, with some mice occasionally running for more than 1 km, ignoring the reward once satiated.
- 2) No signs of distress were visible, from behavioral and local field potential signals, when mice woke up in the treadmill after the craniotomy/probe insertion procedures. Typically, after the anesthesia was turned off, the local field potential started showing quiet sleep associated ripple activity. Stress-related theta activity and signs of struggling were never seen. The first detectable movements were usually occasional

lickings into the lick port, which happened during somnolence/ripple activity. Mice typically started performing the task as soon as they started moving the belt.

- 3) We have actually tried the alternative approach of performing the craniotomy at an earlier stage (half day or one day before) in some other experiments. The down side was that it duplicated the number of time mice were put under anesthesia (once for the craniotomy, once for the probe insertion), which significantly affected behavioral performance in the treadmill. Performing the probe insertion without anesthesia was not ideal either, as the slow process of mice waking-up from anesthesia, useful for shanks stabilization and for confirming CA1 location by ripple activity, was missing.

The following paragraphs were added in the text:

Methods / paragraph 1:

“All experiments were conducted in accordance with institutional regulations (Institutional Animal Care and Use Committees of Korea Institute of Science and Technology), and conformed to the Guide for the Care and Use of Laboratory Animals (NRC 2011).”

Methods / Recording procedures / paragraph 3:

“When mice woke up in the treadmill after the craniotomy/probe insertion procedures, no signs of distress were visible from either behavior or local field potential signals. Behavioral signs of distress, such as mice struggling and grabbing the side posts, were visible only when mice initially experienced head-restriction during training, and were completely absent at any stage of recording sessions. Typically, after the anesthesia was turned off, local field potential progressively started showing quiet sleep associated ripple oscillations. The first detectable movements were usually occasional licks, happening during a period of somnolence/ripple activity. This period was useful for shanks stabilization and for confirming CA1 location by ripple activity. Mice typically started performing the task as soon as they began to move the belt.”

Point #2

In terms of the data, I think the characterization of two different types of place cell responding is interesting, but there is a question-mark over the novelty of this, given the previous report (which they cite) of "landmark vector cells" by Deshmukh and Knierim (2012). A simple way to characterize the findings might be that LV cells are responsive to landmarks in a mostly non-discriminating way (although the rate remapping illustrated by the "identity score" suggests a weak discrimination), while SPV cells are responsive perhaps to olfaction, and thus only fire in one part of the track (otherwise, if olfaction did not play a part, how would the cell know where to fire?). This relatively slight difference could explain why the LV cells responded so quickly to landmarks on the new belt (they were already responding to landmarks, and were not really discriminating them). The slow response of the SPV cells might be because the animal had to learn to detect the odors on the new belt

- that is, it is a product of perceptual learning/attention. Where in the circuit this learning was taking place is an interesting question.

We thank the reviewer for mentioning that the characterization of two different types of place cell responding is interesting. We think that the LV cells we report are indeed equivalent to LV cells described by Deshmukh and Knierim (2012) (accordingly we used the same name). The novelty of our study is 1) the revelation of LV fields as an object-sensory mechanism characterized by instant remapping dynamics, context invariance, and representation of non-spatial information (object identity and saliency), and 2) the anatomical organization of the place field mechanisms. These characteristics were not

investigated in the study of Deshmukh and Knierim (2012). In general, no other study has systematically identified place cell types and tested remapping dynamics, context dependence, and anatomical segregation for the different types separately.

Regarding SPF cells, we agree with the reviewer that they are somehow more specific than LV cells, through a process that involves learning. However, it is unlikely that this specificity comes from conjunction with odors, since in other experiments we carried out (and in Villette et al. 2015), very few cells have place fields in a belt devoid of visual-tactile cues, and the place fields move along with the reward when its position is moved. Instead, for SPF cells in CA3, the mechanism differentiating the duplicated positions of a landmark might be the inputs from CA3 recurrent collaterals, providing sequence learning based information. Considering that cells in both CA3 and CA1s showed single fields, and that recent findings suggest that CA1s might be under CA3 control, we suggest that single fields are implemented in CA3 and then relayed to CA1s. We have considerably revised the discussion along these ideas. The relevant section of the discussion is the following:

Discussion / paragraphs 5 and 6:

“In terms of afferents, CA1d receives most inputs from the region CA2 [22], and may receive stronger inputs from the entorhinal cortex, as CA1d cells show larger entrainment by slow wave oscillations [24]. On the other hand, CA1s pyramidal cells might be relatively more under the control of CA3 inputs, as they show excitatory responses during CA3 stimulations and sharp-wave ripple events, in contrast to CA1d which exhibits mainly inhibitory responses [27]. Interestingly, the underlying mechanism is indirect, involving CA3-to-CA2 feed-forward inhibition [22], causing a lack of excitation in CA1d, and CA1s-to-CA1d feed-forward inhibition [28]. As a result, CA1d and CA1s may belong to two relatively distinct streams of information, namely CA2-CA1d and CA3-CA1s. In this respect, our finding that LV cells are located in CA1d matches recent evidence of strong object influence on CA2 activity [39, 40], and suggest CA2-CA1d as a more ‘sensory’ stream. On the other hand, the CA3-CA1s stream is likely to involve more memory-related mechanisms, because of CA3 extensive recurrent collaterals supporting auto-associative learning [41, 42]. Accordingly, cells in both CA3 and CA1s had mainly single firing fields and slower dynamics. Single fields could not arise from simple visual-tactile sensory mechanisms since every landmark was repeated at least twice on the belt. It is also unlikely that they arise from odors on the belt, since in absence of visual-tactile cues, very few cells have place fields and none retain their position when the reward location is moved [43] (fields correlated either with travel distance [44] or reward distance [43]). Instead, we suggest that single firing fields arise within CA3 from learned conjunctions of sensory, path-integration and local recurrent inputs, before being relayed to CA1s.

Local recurrent inputs are predominant in CA3, and might actually underlie both the uniqueness of firing fields and the fields’ backward drifts that followed landmark manipulations. Indeed, considering the phenomena of spike theta phase precession and spike time dependent plasticity (STDP) [45, 46], an asymmetric input pattern is predicted to emerge through learning, where cells encoding consecutive positions contact each other preferentially in the forward direction [35] (from cells encoding earlier positions to cell encoding latter positions); the reason being that their respective spikes co-occur in theta cycles with delays that are optimal for LTP in the forward direction, while optimal for LTD in the backward direction. As a result, a significant part of the excitation that CA3 cells receive would come from CA3 cells encoding more anterior positions. Such recurrent input pattern would underlie CA3 unique firing fields, as it allows conjunctive representations between ‘animal provenance’ and sensory information. Also, it would provide a natural mechanism for fields’ backward drifts, as fields are expected to shift backward when such input become predominant.”

Point #3

The observation that the different responses were anatomically segregated along the radial axis of CA1 is also interesting, as is the way in which the cells responded to the different types of cues, and the fact that cells did not change their response patterns when the cues changed (with the second belt). However I would first like to see some more robust statistics to support the claim of anatomical segregation. The procedure of calculating pairwise distances between cells is flawed because it multiplies the data without justification - the correct comparison, to claim that the LV cells were deeper than the SPF cells, would be to calculate the depth of the two groups and then compare with a straightforward unpaired two-tailed (not one-tailed) t-test.

As suggested by the referee, we have done an additional analysis to quantify the cells anatomical segregation along the radial axis. A new figure panel (fig 2D) was added and the relevant paragraph was modified:

Results / paragraph 3:

“Then, since each shank likely reached a different depth of the CA1 pyramidal layer, we estimated for each shank the position of the recording site with maximum ripple power, and expressed cell depths in terms of distance from that position [24]. We observed that LV cells were concentrated in a deeper part of the layer than SPF cells, as LV were located on average $4.4 \pm 2.8 \mu\text{m}$ above ripple peak position while SPF cells reside on average $-8.2 \pm 3.2 \mu\text{m}$ below (Fig. 2D, $P < 0.01$, unpaired t-test).”

Point #4

The spatial relationship of the LV fields to the landmarks raises some interesting questions about the mechanism and/or function.

We agree with the reviewer, and it will be interesting in the future to test further the underlying mechanisms. At this point, we speculate that LV cells originate from an interaction of grid cells/border cells inputs from MEC and ‘object’ inputs from LEC, contributing respectively the spatial and non-spatial aspects. Such independent origins would fit our observation that the spatial relationship was maintained across different belts with distinct type of objects. The relevant paragraph was modified as follow:

Discussion / paragraph 3:

“Our results provide new insights on the mechanisms underlying LV cells. LV firing fields were closely associated with sensory mechanisms, as they showed repetitions and instant dynamics, but encoded both spatial (landmark distance) and non-spatial information (landmark identity and saliency). While non-spatial object information is believed to reach the hippocampus via LEC [14, 15], spatial information might come from MEC inputs. Hence, a possible scenario is that LV cell activity emerges from an interaction of LEC and MEC inputs, contributing respectively the ‘landmark specificity’ and ‘landmark distance’ aspects. For instance, considering that grid cells reset and activate at similar distances in the repeated alleys of a ‘hairpin’ maze [47], and that border cells activate near boundaries [18], it is possible that some combinations of grid cells and border cells encode particular positions near the landmarks, supplying the hippocampus with discrete spatial inputs, which sum with the object specific (but less spatially tuned [14, 15]) inputs from LEC. Along these lines, the fact that spatial relationships were preserved across belt environments would suggest that each LV cell remains strongly connected to a particular subset of MEC inputs, and the immediate emergence of spatial relationships upon novel landmarks additions would suggest a pre-configuration of underlying connectivity patterns.”

We also suggest some functional implications in the concluding paragraph, which was modified as follow:

Discussion / paragraph 7:

“Our findings suggest a functional division between CA1 deep and superficial layers. While LV cells in the deeper layer supply sensory mediated representation of self-position and object locations, cells with looser ties to landmarks tackle spatial representation on a more global level, using both sensory and memory information, and may also be more flexible for integrating non-spatial factors such as reward, goal and time [7,9,10]. The coexistence of these distinct place field mechanisms suggests that diverse types of spatial associations, involving distinct levels of specificity, precision, and portability across environments, might occur in parallel. Future experiments using selective inactivation of deep and superficial CA1 cell groups should help reveal their relative contribution to memory.”

Other comments

R1: P4 What does "quasi absent" mean? Also, "robust across distinct belt environments" is not understandable at this point.

The reviewer is correct. The sentence was changed as follow:

Introduction / paragraph 3:

“We extend these findings by showing that, in contrast to SPF cells, 1) LV cells are more frequent in CA1 than in CA3 by an order of magnitude, 2) LV cells are concentrated in the deep portion of CA1 pyramidal layer, 3) LV cells activity is sustained across distinct environments, and 4) LV cells show instantaneous field dynamics in response to object manipulation.”

R1: P5 The description of landmarks at this stage should make clear that the landmarks were on the belt and not fixed in the room (it took me a while to appreciate this point).

As suggested by the reviewer, we have added a sentence to make this point clearer:

Results / paragraph 1:

“... The landmarks were fixed on the belt and spanned over the 5 cm width of the belt, providing visual-tactile stimulation to both sides of the mice.”

R1: A little later in that paragraph, it should be explained that the LV cells usually responded to all the landmarks regardless of type (though there was, as they show, some rate remapping).

We would rather not state too strongly that “LV cells usually responded to all landmarks”, as this was true only when all landmarks were salient. At this point, we would rather be less specific, and address this issue later when we analyze the impact of landmark identity and saliency. Hence, we have modified the text in a slightly different manner than suggested by the reviewer:

Results / paragraph 1:

“... and, cells that exhibited firing fields tightly coupled to the landmarks on the belt, repeating in similar fashion at multiple landmark positions, in several cases regardless of landmark types (Fig.1C, 1D, See Methods).”

R1: What was the reason for having three different lengths of belt?

In the experiment where the same cells were recorded in two different belts, we used different belt lengths (190cm and 140cm) to increase the difference between the two belt environments. Otherwise, the length of the belts varied between 170 to 230 cm, which was essentially determined by the number of cues on the belt.

R1: Was the distribution of LV and SPF cells different between CA3 and CA1 (by chi-square)?

Yes, we thank the referee for this suggestion. We have added the information in the text:

Results / paragraph 2:

“The distributions were significantly different ($p = 0$, Chi-square null hypothesis of independence, Chi-square = 119.7, degrees of freedom: $k = 2$).”

R1: P6 The "identity index" was unintuitive to me. A more natural (to my way of thinking) measure would have been the ratio of the peak of the best landmark to the peak of the secondary one; or a discrimination index (difference in the peaks divided by the sum). What was the reasoning for the measure used?

The reasoning was the following:

- 1) To say that a cell prefers a given cue based on its type, the two fields associated with that cue should be larger than any of the other fields. This is the reason to compare the smallest field of the dominant cue with the largest field of the other cue. If it is larger, the condition is met, and our method returns an index > 0 (otherwise identity index < 0). Using the ratio of the peak of the best landmark to the peak of the secondary one (or the discrimination index) would not capture this aspect.
- 2) Additional information we wanted to be represented in the index is how different is the firing rate for the two kinds of landmarks. We used a subtraction (after normalization) since it is a simple measure that will capture this information and give a number between -1 and 1. Values closer to 1 correspond to larger size difference between fields of the two landmarks.

We have modified the text the following manner to make the method clearer:

Results / paragraph 5:

“... We implemented an identity index, which captured 1) whether all fields of the dominant landmark had higher peak rates than any fields of the secondary landmark, and 2) the difference in field peak rates between the two landmarks. Specifically, the identity index was computed as the difference, after normalization, in peak firing rates between dominant and secondary landmarks, considering only the smallest field of the dominant landmark and the largest field of the secondary landmark (Fig. 3B). An index value above zero indicates that all fields encoding the dominant landmark have higher peak rates than any of the fields encoding the secondary landmark. Large index values (close to 1) correspond to large rate differences between the two landmarks”

Reviewer #2 (Remarks to the Author)

The finding of this anatomical segregation is novel but the weakness of this work is that the origins or mechanisms are not explored or tested. This is where my enthusiasm for the findings was decreased. The authors favor the interpretation that different anatomical

projections to CA1 that each represent different types of information underlie their findings. However, they do not test the influence of different inputs on CA1 activity by incorporating inactivations, lesions, or manipulations in mutant mice where individual inputs can be manipulated. Even without these additional experiments the manuscript would be stronger with an improved discussion of the anatomy and a presentation for a working hypothesis on how the varied inputs could result in their findings. Currently the authors remain non committed suggesting a number of potential contributions or theories that are not well discussed.

We thank the Reviewer for mentioning that our finding about the anatomical segregation is novel and certainly share the reviewer's wish for mechanistic insights on the anatomical segregation. We feel that identifying the origins or mechanisms is beyond the scope of this manuscript and constitute a separate follow-up project that would take possibly years to conduct. In fact, we plan to extend our follow-up research along these lines. Therefore, we decided to focus on the Reviewer's alternative suggestion; *"Even without these additional experiments the manuscript would be stronger with an improved discussion of the anatomy and a presentation for a working hypothesis on how the varied inputs could result in their findings"*.

In this revision, we have now extended the discussion with possible working hypotheses. To summarize, we found that our findings fit quite nicely the picture emerging from recent studies (in particular the critical study outlined by the reviewer in comment #5), that CA3-CA1s and CA2-CA1d constitute distinct streams of information. Indeed, matching a CA3-CA1s stream, we found that cells in CA3 and CA1s showed single firing fields, slower dynamics, and backward drifts. We discuss how these phenomena could originate from CA3 recurrent network dynamics. On the other hand, objects are reported to influence CA2 activity in particular, and LV cells are seen in CA1d, consistent with a more 'sensory' CA2-CA1d stream. As suggested by the reviewer, we have revised the discussion to incorporate these ideas.

Please refer to "Discussion / paragraphs 5 and 6" in the response to Point#2 of Reviewer#1

- 1) The authors argue that the LEC and MEC inputs could potentially support the LV and SPF cells differentially. How do the authors then explain their findings in Figure 2 in which the proportion of LV and SPF cells does not change across the proximal-distal axis of CA1 where LEC and MEC inputs are known to be differentially segregated?

Actually, we do not say that LEC and MEC inputs support LV and SPF cells differentially, but instead, that the spatial and non-spatial aspects of LV cells might originate from MEC and LEC respectively. We apologize for the confused formulation in the previous version of the paragraph. The paragraph was modified accordingly.

Please refer to "Discussion / paragraph 3" in the response to Point#4 of Reviewer#1

- 2) From the 3D anatomy reconstruction in Supp Figure 2 it appears that the majority of CA1 recordings are located in the distal region of CA1 closer to subiculum and few red lines appear to be located in the distal areas closer to CA2. How representative is the example shown. There should be a summary for the distal and proximal axis across animals in order to see the extent to which the total region was sampled across animals.

We thank the reviewer for calling attention to this important point. The reconstruction in Supplementary Figure 2 is only representative of a subset of recordings. In more than half of the recordings, the shanks of the silicon probes were actually oriented parallel to the proximo-distal axis. We have added to the figure (See supplementary Figure 2E) examples

of coronal sections where shanks were oriented along the proximo-distal axis. Also, as suggested by the reviewer, we have incorporated a figure that summarizes shanks locations/orientation for all animals on a 2D layout of CA1 (See Supplementary Figure 2F).

3) The authors need to show to what extent the proportion of cells they report across experiments is driven by individual mice or by anatomical segregation. For instance can the proportion of drifting vs. switching cells be explained by individual mice in which one mouse had more drifting cells and another more switching cells. For all experiments numbers need to be reported or data shown that verify that the phenomenon they describe and the proportions of cells they report hold within individual mice.

The reviewer is correct that the manuscript was not clear about this subject. Two figures were added in the supplementary material. Supplementary Figure 4H provides the numbers of drifting and switching cells recorded in each mouse, for experiments where cues were manipulated. Supplementary Figure 2G shows the number of LV and SPF cells recorded by each silicon probe illustrated in Supplementary Figure 2F.

4) Can the authors clarify in the methods whether the speed of the track is controlled by the experimenter and is continuous at a set speed, or whether the animal can modulate this by their own behavior.

We thank the Reviewer as we forgot to mention this. As suggested by the Reviewer, we have added the following sentence in the result and in the method sections:

Results / paragraph 1 and Methods / Behavioral training:

“The treadmill was not motorized, but consisted of a light velvet belt laying on two 3D printed wheels, which mice moved themselves at will [30]“

5) the authors should discuss their findings and views of the circuits with the new data that has been recently reported describing activity in CA1d and CA1v during SWR activity (de la Prida).

We thank the Reviewer for this suggestion. As mentioned before, this important reference is now a major point of the discussion (please refer to “Discussion / paragraphs 5 and 6” in the response to Point#2 of Reviewer#1). The reference is also referred to in the introduction as follow:

Introduction / paragraph 2:

“Also, CA1d and CA1s cells show distinct phase relation with theta/gamma oscillations [24], are differentially affected by reward signals [25], and activate in sequence during ripple oscillations in hippocampal slice [26] while showing respectively inhibitory and excitatory responses during ripple oscillations in vivo [27].”

REVIEWERS' COMMENTS:

Reviewer #1 (Remarks to the Author):

The manuscript is considerably improved, and in particular the extra details about the ethical considerations have allayed my concerns about the conduct of the experiment.

The basic finding reported in the paper is of interest and will shape thinking about the organization of hippocampal circuitry. I have a number of other minor comments, some of which are detailed below and the remainder of which are in the annotated pdf.

First, the title is slightly misleading – “increasingly” has connotations of progressive, gradual change and implies a gradual shifting of control by landmarks, but really, there was a binary switch between one type of cue control and another. It might be better to say something like “Stronger landmark control of place fields in deep than in superficial CA1” or something less gradual-sounding.

Second, the numerous grammatical errors remain. Most of these do not change the meaning, but they are annoying. I have corrected many of them by annotating the pdf but this ideally is not the job of a reviewer!

Third, I am uneasy about the nomenclature of these cells. The term “landmark vector cell” carries a functional assumption – which could well be warranted, and it is a term that has been used before, so I agree with using it here too. However the other cells are called “single place field cells” and this term is purely phenomenologically descriptive, and might not pertain to all environments. Furthermore, wouldn't some LV cells also have single place fields in the right conditions? I wonder if it would be better to call them “context modulated cells” or something that also better describes their firing correlates, and makes it easier for the reader to remember the important differences between the cell types.

Other things:

The landmarks need better describing – they are hard for the reader to visualize. It's not quite clear what happened with the vertical tubes – did the animals run through these, or were they lined along the edges, or what? The figures suggest the former but this could be clarified in the text.

Statistics – these are under-reported in several places: the t statistic and degrees of freedom should be reported for t-tests, etc.

Discussion – I would suggest to open this with the main finding, which is the anatomical segregation of the different types of responding. I found the large amount of added text a bit rambling and speculative, and would suggest to remove or greatly shorten it, while retaining the final paragraph which is a good, succinct summary.

Reviewer #2 (Remarks to the Author):

The authors have responded to all reviewer comments. Additional analyzes did not change the main findings presented in the original submission. The work by Geiller et al. is an incremental study that builds upon the description of landmark vector (LV) cells which were first reported by the Knierim lab in 2013 in the journal *Hippocampus*. The additional figure panels added in response to reviewer critiques further support their methods and outcomes, which describe that distinct functional cell classes (LV and Single Place Fields) are selectively distributed across the deep and superficial layers of CA1. The main criticism of the revised manuscript is the readability of the Discussion. At present this section of the manuscript is difficult to read and not well organized.

1) In the absence of causal data the authors speculate on the origins of the anatomical segregation of the LV and SPF cells. Previous critiques asked the authors to focus on a primary hypothesis on the anatomical segregation in their discussion as a laundry list of possibilities were provided for the occurrence of this phenomenon with no arguments in favor or against. It appears that the hypothesis is centered on the differential influence of CA2 and CA3 inputs to CA1d and CA1s. The discussion should be reorganized to better present this reasoning. As currently written, the authors first argue that the inputs from LEC and MEC support the anatomical distribution of LV and SPF cells although these inputs are segregated across the proximal/distal axis of CA1 in which the authors do not find a difference (this was not mentioned). The authors then proceed to argue in newly added text that it is the CA2 and CA3 inputs that underlie the reported segregation. The new arguments for a specific role for CA2 and CA3 to their findings does improve the manuscript, but these ideas need to be better integrated into the discussion.

2) The first paragraph of the discussion lists possible reasons for differences between the Deshmuck et al., 2013 paper in which LV cells were originally described and the findings reported in this submission. One obvious possibility, that was not discussed, is a potential difference between place field properties in mice and rats. The Knierim lab performed all studies in rats, the Royer lab mice. It has been shown from the Kentros lab that properties of hippocampal place fields in mice are quantitatively different than those in rats. Furthermore, it has also been shown by comparing work from the Allen lab and the Moser lab the grid cell properties in MEC are also different between rats and mice. The contribution of this submission is the confirmation of LV cells in the CA1 of mice, the segregated anatomical position of these cells in CA1d, and a more in depth description of their firing properties. Knowing that the response properties of place fields has already been shown to differ in some ways between rat and mouse strains it should be a major consideration when comparing data with the Deshmuck study.

3) There are no statements about the functional significance of the anatomical segregation of LV cells to CA1d. As is the discussion is largely descriptive.

Point-by-point response

Reviewer #1 (Remarks to the Author):

The manuscript is considerably improved, and in particular the extra details about the ethical considerations have allayed my concerns about the conduct of the experiment. The basic finding reported in the paper is of interest and will shape thinking about the organization of hippocampal circuitry. I have a number of other minor comments, some of which are detailed below and the remainder of which are in the annotated pdf.

Point #1

First, the title is slightly misleading – “increasingly” has connotations of progressive, gradual change and implies a gradual shifting of control by landmarks, but really, there was a binary switch between one type of cue control and another. It might be better to say something like “Stronger landmark control of place fields in deep than in superficial CA1” or something less gradual-sounding.

We agree with the reviewer that our data do not demonstrate a gradual shift, and that the title is misleading in that respect. We have modified accordingly the title and the abstract.

Title: “Place cells are more strongly tied to landmarks in deep than in superficial CA1”

Abstract (2nd sentence): “Here we show that, in mice running on a treadmill enriched with visual-tactile landmarks, place cells are more strongly controlled by landmark-associated sensory inputs in deeper regions of CA1 pyramidal layer (CA1d)”

Point #2

Second, the numerous grammatical errors remain. Most of these do not change the meaning, but they are annoying. I have corrected many of them by annotating the pdf but this ideally is not the job of a reviewer!

We apologise for the remaining grammatical errors and thanks the reviewer for reporting them.

Point #3

Third, I am uneasy about the nomenclature of these cells. The term “landmark vector cell” carries a functional assumption – which could well be warranted, and it is a term that has been used before, so I agree with using it here too. However the other cells are called “single place field cells” and this term is purely phenomenologically descriptive, and might not pertain to all environments. Furthermore, wouldn’t some LV cells also have single place fields in the right conditions? I wonder if it would be better to call them “context modulated cells” or something that also better describes their firing correlates, and makes it easier for the reader to remember the important differences between the cell types.

We agree with the reviewer that the label “context-modulated cells” is better capturing the mechanism underlying “single place field cells” (which was a purely phenomenological term we chose because it was non-committing in terms of mechanism), in addition to fit our finding that these cells were more specific to context. We have renamed the cells accordingly. We thank the reviewer for this suggestion.

Point#4

The landmarks need better describing – they are hard for the reader to visualize. It's not quite clear what happened with the vertical tubes – did the animals run through these, or were they lined along the edges, or what? The figures suggest the former but this could be clarified in the text.

The erected landmarks such as the spines and the vertical tubes were indeed lined along the edges of the belt, and in addition were flexible. The mice ran through these without interruptions. As suggested by the reviewer, we have revised the text to make this point clearer. The relevant sentence read as follows: (Results paragraph 1) "The landmarks were fixed on the belt and were composed of vertically erected flexible objects or horizontally laid objects, lined along the edges of the belt, providing visual-tactile stimulation to both sides of the mice without interfering with their locomotion."

Point#5

Statistics – these are under-reported in several places: the t statistic and degrees of freedom should be reported for t-tests, etc.

For all statistical tests reported in the text and figure legends, we have added the n, and for t-tests we have added the degrees of freedom in addition to the n.

Point#6

Discussion – I would suggest to open this with the main finding, which is the anatomical segregation of the different types of responding. I found the large amount of added text a bit rambling and speculative, and would suggest to remove or greatly shorten it, while retaining the final paragraph which is a good, succinct summary.

As suggested by the reviewer, we have reorganised the discussion and shorten or removed some of the speculative sections. Now the discussion opens with the paragraph on the anatomical segregation of cell types along CA1 radial axis. We have indicated using 'comments' in the text the sections that were removed, shortened, or added. We thank the referee for this suggestion.

Reviewer #2 (Remarks to the Author):

The authors have responded to all reviewer comments. Additional analyses did not change the main findings presented in the original submission. The work by Geiller et al. is an incremental study that builds upon the description of landmark vector (LV) cells which were first reported by the Knierim lab in 2013 in the journal *Hippocampus*. The additional figure panels added in response to reviewer critiques further support their methods and outcomes, which describe that distinct functional cell classes (LV and Single Place Fields) are selectively distributed across the deep and superficial layers of CA1. The main criticism of the revised manuscript is the readability of the Discussion. At present this section of the manuscript is difficult to read and not well organized.

Point #1

1) In the absence of causal data the authors speculate on the origins of the anatomical segregation of the LV and SPF cells. Previous critiques asked the authors to focus on

a primary hypothesis on the anatomical segregation in their discussion as a laundry list of possibilities were provided for the occurrence of this phenomenon with no arguments in favor or against. It appears that the hypothesis is centered on the differential influence of CA2 and CA3 inputs to CA1d and CA1s. The discussion should be reorganized to better present this reasoning. As currently written, the authors first argue that the inputs from LEC and MEC support the anatomical distribution of LV and SPF cells although these inputs are segregated across the proximal/distal axis of CA1 in which the authors do not find a difference (this was not mentioned). The authors then proceed to argue in newly added text that it is the CA2 and CA3 inputs that underlie the reported segregation. The new arguments for a specific role for CA2 and CA3 to their findings does improve the manuscript, but these ideas need to be better integrated into the discussion.

As suggested by both reviewers, we have reorganised the discussion to improve its readability and increase the emphasis on the anatomical segregation of functions, which is the main finding. Now the discussion open with the paragraph on the anatomical segregation of cell types along CA1 radial axis. The next paragraph is on the discrepancy of LV cells number between our study and a previous study from the Knierim lab. The next three paragraphs address the possible mechanisms underlying each cell type (LV cells, switching cells and drifting cells). The last paragraph speculates on the functional significance of our findings.

Point #2

2) The first paragraph of the discussion lists possible reasons for differences between the Deshmuck et al., 2013 paper in which LV cells were originally described and the findings reported in this submission. One obvious possibility, that was not discussed, is a potential difference between place field properties in mice and rats. The Knierim lab performed all studies in rats, the Royer lab mice. It has been shown from the Kentros lab that properties of hippocampal place fields in mice are quantitatively different that those in rats. Furthermore, it has also been shown by comparing work from the Allen lab and the Moser lab the grid cell properties in MEC are also different between rats and mice. The contribution of this submission is the confirmation of LV cells in the CA1 of mice, the segregated anatomical position of these cells in CA1d, and a more in depth description of their firing properties. Knowing that the response properties of place fields has already been shown to differ in some ways between rat and mouse strains it should be a major consideration when comparing data with the Deshmuck study.

We thank the reviewer for raising this possibility. We have included this idea and a relevant new reference in the appropriate section of the discussion: (Discussion paragraph 2) "Last, it is possible that the quantitative discrepancy between the two studies reflects differences between mouse and rat species, since rats were used in the study of Deshmuck et al. Compared to rats, mice might use more the local cues over the distal cues [45], resulting in a larger number of LV cells in mice."

Point #3

3) There are no statements about the functional significance of the anatomical segregation of LV cells to CA1d. As is the discussion is largely descriptive.

The reviewer is correct that the functional significance of the anatomical segregation was not extensively explored, as the emphasis was put on the mechanisms. We have now extended the last paragraph, which already addressed functional significance in relatively general terms, with

additional speculations taking into consideration the difference in brain regions targeted by CA1d and CA1s. The relevant section read as follow:

“The coexistence of these distinct place field mechanisms suggests that diverse types of spatial associations, involving distinct levels of specificity, precision, and portability across environments, might occur in parallel. The fact that CA1d generates most CA1 projections to brain regions involved in goal oriented behaviors (ventral striatum/nucleus accumbens, septal area and orbitofrontal cortex)[23] might underlie a behavioral benefit for reward prediction mechanisms to be linked with discrete cues and transferable across environments, while CA1s predominant feedback projections to the medial temporal cortex[23] might contribute the contextual information to episodic memory processes in this region.”

We thank both reviewers for their constructive comments.